# SpheriQ: Probabilistic Hyperbolic Reasoning for Interpretable Recommendation

## Abstract

Recommendation models are increasingly used in settings where ambiguity and transparency matter, yet many approaches are deterministic or poorly calibrated. We present SpheriQ, a geometric framework that embeds users, items, and tags as probabilistic regions in hyperbolic space. The centre encodes the semantic position while the radii capture predictive uncertainty; a Gaussian semantic kernel on the manifold enables calibrated, transitive composition along the user to tag to item paths. This bridges symbolic and distance based paradigms, providing concept level traces and confidence estimates with the efficiency of lightweight embeddings. We instantiate SpheriQ with automatic tag construction and Riemannian optimisation, and evaluate it on news, books, and commonsense reasoning benchmarks. Across datasets, our model pairs strong ranking performance with improved calibration and semantic diversity, while remaining efficient to train. The concrete goal is to unify ranking, calibration, and explanation by casting recommendation as probabilistic concept level reasoning in hyperbolic space, with tags serving as semantic pivots rather than end objectives.

## 1 Introduction

Recommender systems shape how users engage with news, books, and knowledge online. Embedding based and neural recommenders (He et al., 2020; Ong et al., 2021) have achieved notable progress, yet two challenges persist. First, most models are *deterministic*: they return point predictions without calibrated confidence, leading to overconfident behaviour in safety critical domains such as healthcare or education (Mazurowski, 2013; Mesas & Bellogín, 2017). Recent work on calibrated recommendation highlights this issue (Naghiaei et al., 2022). Second, such systems are largely *opaque*, offering little insight into their internal reasoning, which undermines trust in domains requiring accountability.

Tags and semantic attributes provide a natural bridge between users and items (Zhang et al., 2010; Chen et al., 2020), and hyperbolic geometry is effective for modelling hierarchical structure (Nickel & Kiela, 2017). However, existing tag based models rely on deterministic diffusion or tripartite reasoning, while hyperbolic methods typically ignore uncertainty. Explainable recommendation and symbolic reasoning (Wilson et al., 2014; Dong et al., 2025) show the value of semantic alignment, and knowledge graph or rule based methods highlight structured explanations (Ai et al., 2018; Ma et al., 2019), yet uncertainty, interpretability, and semantics are usually treated as separate goals.

Probabilistic methods such as VAEs and Bayesian matrix factorisation capture predictive distributions (Wang et al., 2023) but remain *Euclidean and opaque*, lacking explicit semantic reasoning. Recent probabilistic embeddings in hyperbolic or Gaussian spaces (Nickel & Kiela, 2017) model uncertainty geometrically but do not address recommendation or concept level explanations. Other work explores uncertainty in implicit feedback and diversity (Coscrato & Bridge, 2022; Liu et al., 2019), but without unifying calibration and explanation.

**Aim.** We propose **SpheriQ**, a probabilistic hyperbolic reasoning framework that unifies ranking accuracy, calibration, and explanation. Users, items, and tags are embedded as *probabilistic spheres*: centres encode semantic position and radii capture predictive uncertainty. A Gaussian semantic kernel enables *soft transitive reasoning* along user–tag–item paths, yielding calibrated scores and faithful rationales. This formulation generalises symbolic and distance based paradigms while providing a mathematically grounded mechanism for trustworthy recommendation.

Our contributions are threefold:

- A **probabilistic hyperbolic embedding framework** that represents users, items, and tags as spheres with uncertainty radii, enabling calibrated reasoning beyond deterministic embeddings.
- A **semantic kernel for probabilistic transitivity**, composing user–concept–item similarities into interpretable, uncertainty aware recommendations.
- A **theoretical link between geometry and logic**, showing that sphere inclusions correspond to entailments. Empirical results on MIND, GoodBooks, BookCrossing, and Avicenna show up to 8.3% higher NDCG@10 and 6.7% lower calibration error over strong baselines, with training efficiency.

By unifying probabilistic geometry with semantic reasoning, SpheriQ provides an interpretable and uncertainty aware foundation for recommendation, advancing beyond tag diffusion and deterministic hyperbolic models toward *trustworthy systems* that balance accuracy, transparency, and calibration.

## 2 PRELIMINARIES

This section introduces the mathematical background and data processing underpinning SpheriQ. These components are not themselves contributions but provide the foundation upon which the method is built.

### 2.1 NOTATION

Let $\mathcal{U}$, $\mathcal{I}$, and $\mathcal{T}$ denote the sets of users, items, and tags, respectively. Each entity $i$ is represented as $O_i = (\mu_i, \sigma_i^2)$, where $\mu_i$ is the embedding centre and $\sigma_i^2$ its associated variance. Vectors are initially embedded in $\mathbb{R}^n$. The hyperbolic space is modelled as the $n$-dimensional Poincaré ball

$$\mathbb{D}^n = \{x \in \mathbb{R}^n : \|x\| < 1\}.$$

An encoder $E(\cdot)$ maps raw text or metadata into $\mathbb{R}^n$.

### 2.2 HYPERBOLIC GEOMETRY

The Poincaré ball induces the distance between two points $\mu_i, \mu_j \in \mathbb{D}^n$ as

$$d_{\mathbb{D}}(\mu_i, \mu_j) = \cosh^{-1}\left(1 + 2\frac{\|\mu_i - \mu_j\|^2}{(1 - \|\mu_i\|^2)(1 - \|\mu_j\|^2)}\right). \tag{1}$$

To initialise embeddings, encoder outputs are projected onto the manifold via the exponential map at the origin:

$$\mu_i = \exp_0\left(\frac{E(x_i)}{\|E(x_i)\|} \tanh(\|E(x_i)\|)\right). \tag{2}$$

This ensures that all representations lie within $\mathbb{D}^n$.

### 2.3 PROBABILISTIC SPHERE EMBEDDINGS

Each entity is represented as a *probabilistic sphere*

$$O_i = (\mu_i, \sigma_i^2), \qquad \mu_i \in \mathbb{D}^n, \ \sigma_i^2 \in \mathbb{R}_+. \tag{3}$$

Uncertainty is parameterised via a softplus transform:

$$\sigma_i^2 = \log\big(1 + \exp(w_i)\big), \qquad w_i \in \mathbb{R} \text{ trainable}, \tag{4}$$

which guarantees positivity and smooth gradients. We assume isotropic variance for tractability. Intuitively, small radii indicate confident semantics, whereas large radii reflect predictive uncertainty.

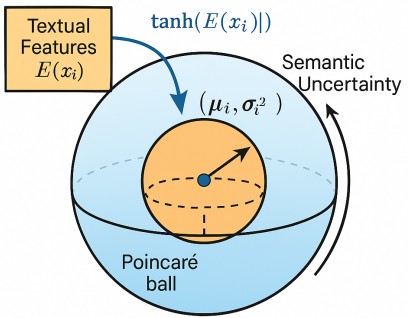 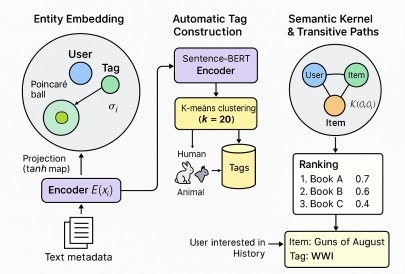

Figure 1: **SpheriQ overview.** *Left*: an entity as a probabilistic sphere with centre $\mu_i$ and radius $\sigma_i$ in the Poincaré ball. *Right*: pipeline with encoder initialisation, lightweight tag construction, and the semantic kernel used for transitive reasoning along user→tag→item paths.

## 2.4 SEMANTIC KERNEL

The similarity between two spheres $O_i$ and $O_j$ is defined by a Gaussian kernel in hyperbolic space:

$$K(O_i, O_j) = \exp\left(-\frac{d_{\mathbb{D}}(\mu_i, \mu_j)^2}{2\lambda(\sigma_i^2 + \sigma_j^2)}\right), \tag{5}$$

which generalises the radial basis function to curved geometry while explicitly incorporating uncertainty. The temperature parameter $\lambda > 0$ controls kernel sharpness; unless otherwise stated we fix $\lambda = 1$. Key properties such as boundedness, symmetry, and monotonicity in variance are established in the appendixD.

## 2.5 TAG SETS

For each dataset $d \in \{\text{MIND, GoodBooks, Book\_Crossing, Avicenna}\}$ we construct a tag set $\mathcal{T}_d$ using lightweight, dataset-specific procedures:

- **MIND**: publisher-provided categories and subtopics;
- **GoodBooks** and **Book\_Crossing**: Sentence-BERT (`all_MiniLM_L6_v2`) embeddings clustered via K-means ($k = 20$ chosen by Silhouette analysis), with each item assigned to its nearest centroid;
- **Avicenna**: symbolic concepts extracted from logical triples.

All tag sets are precomputed and fixed for reproducibility; sensitivity to $k$ is reported in ablations.

Each tag $t \in \mathcal{T}_d$ is then embedded as a probabilistic sphere $O_t = (\mu_t, \sigma_t^2)$ in the same Poincaré space as users and items. For clustered tags, $\mu_t$ is initialised from the centroid embedding; for symbolic tags, from the corresponding concept encoder. This alignment ensures that users, items, and tags share a unified geometric space, allowing semantic kernels $K(O_u, O_t)$ and $K(O_t, O_j)$ to be consistently composed in transitive reasoning.

## 3 METHOD

We cast recommendation as probabilistic reasoning over user, tag, and item spheres in hyperbolic space. SpheriQ composes semantic similarities along concept paths and learns to rank items while calibrating uncertainty.

### 3.1 PROBLEM STATEMENT

Given user, item, and tag sets $\mathcal{U}, \mathcal{I}, \mathcal{T}$ and their probabilistic sphere representations $O_u, O_j, O_t$ (Sec. §2), our goal is to produce, for each user $u$, a calibrated ranking over items $j \in \mathcal{I}$ together with an explicit semantic explanation via a tag $t \in \mathcal{T}$.

## 3.2 TRANSITIVE SCORING

We score user–item pairs by composing similarities along user–tag–item chains:

$$s(u,j) = \max_{t \in \mathcal{T}} f\big(K(O_u, O_t),\, K(O_t, O_j)\big), \tag{6}$$

where the *fuzzy composition* aggregates the two kernel factors:

$$f_{\text{prod}}(a, b) = ab, \qquad f_{\min}(a, b) = \min(a, b). \tag{7}$$

The outer maximum selects the most confident conceptual path and exposes an explanation $u \rightarrow t^\star \rightarrow j$, where $t^\star = \arg\max_{t \in \mathcal{T}} f(K(O_u, O_t), K(O_t, O_j))$. For efficiency we optionally prefilter per user to the top $k$ tags by $K(O_u, O_t)$, reducing the per-pair cost to $O(k)$.

**Inference.** At test time we compute $s(u, j)$ for all candidate items $j$ and return the top-$k$ items, each accompanied by its maximising tag $t^\star$ as an explicit rationale.

**Practical note (smooth training).** For stability, a *soft* surrogate can replace the hard max during training:

$$s_\tau(u, j) = \tau \log \sum_{t \in \mathcal{T}} \exp\Big(\tfrac{1}{\tau} f(K(O_u, O_t), K(O_t, O_j))\Big),$$

with temperature $\tau \downarrow 0$ recovering Eq. equation 6. We keep the hard max at inference.

## 3.3 LEARNING OBJECTIVE

We optimise a contrastive (softmax) ranking loss over observed positives and sampled negatives:

$$\mathcal{L}_{\text{rank}} = - \sum_{(u, j^+) \in \mathcal{D}} \log \frac{\exp(s(u, j^+))}{\exp(s(u, j^+)) + \sum_{j^- \in \mathcal{N}(u)} \exp(s(u, j^-))}. \tag{8}$$

Geometry and uncertainty are regularised via

$$\mathcal{L}_{\text{reg}} = \sum_{i} \Big( \lambda_1 \|\mu_i\|^2 + \lambda_2 \, \text{KL}\big(\mathcal{N}(0, \sigma_i^2) \,\|\, \mathcal{N}(0, \sigma_0^2)\big) \Big), \tag{9}$$

with prior variance $\sigma_0^2$. The full objective is

$$\mathcal{L} = \mathcal{L}_{\text{rank}} + \mathcal{L}_{\text{reg}}. \tag{10}$$

**Training protocol.** We train on mini-batches of observed interactions, sampling 100 negatives uniformly per positive; evaluation ranks over the full item set. This yields unbiased gradients while scaling to large vocabularies.

## 3.4 OPTIMISATION AND COMPLEXITY

Parameters are updated with Riemannian Adam (Kochurov et al., 2020) using Möbius operations to maintain feasibility in $\mathbb{D}^n$. Gradients are computed in tangent spaces, updated, and mapped back via the exponential map. We cache hyperbolic distances and kernel scores where safe to do so.

**Complexity and memory.** With exact scoring, a full epoch costs $O(|E| \cdot |\mathcal{T}|)$; with per-user prefiltering it is $O(|E| \cdot k)$, where $|E|$ is the number of observed interactions. Memory is dominated by embeddings of users, items, and tags, i.e., $O((|\mathcal{U}| + |\mathcal{I}| + |\mathcal{T}|)\, n)$, plus optional kernel caches.

## 3.5 PROPERTIES USED IN PRACTICE

We rely on two properties:

- **Monotonicity in uncertainty.** For fixed centres, $K(O_i, O_j)$ decreases as $\sigma_i^2 + \sigma_j^2$ increases; thus the composed score $s(u, j)$ is non-increasing in $\sigma_u^2, \sigma_t^2, \sigma_j^2$. This underpins our calibration analyses.
- **Path selection.** For finite $\mathcal{T}$, the maximum in Eq. equation 6 is attained by some $t^\star$, yielding an explicit explanation certificate $u \rightarrow t^\star \rightarrow j$.

**Kernel invariances.** The kernel in Eq. equation 5 is bounded, symmetric, and invariant to Poincaré isometries, so equivalent geometric embeddings induce identical recommendations.

### 3.6 IMPLEMENTATION NOTES

Centres are initialised with encoders $E(\cdot)$ and fine-tuned end-to-end. Unless stated, $\lambda = 1$ and $f = f_{\text{prod}}$ (we report $f_{\min}$ in ablations). Tag sets are precomputed for reproducibility. Because scoring necessarily factors through a tag, each prediction is accompanied by an interpretable semantic certificate $u \to t^\star \to j$.

## 4 EXPERIMENTAL SETUP

### 4.1 DATASETS

We evaluate **SpheriQ** on four publicly available datasets spanning news recommendation, book preference modelling, and commonsense reasoning. (1) **MIND-small** (Wu et al., 2020)[1] is a large-scale news recommendation benchmark with click logs and article metadata; publisher-provided topics are used as semantic tags. (2) **GoodBooks-10K** (Zajac, 2017)[2] contains implicit feedback ratings on 10,000 books, where Goodreads genres and tags act as conceptual anchors. (3) **Book-Crossing** (Ziegler et al., 2005)[3] is a sparse user–book interaction dataset; we cluster metadata into broad semantic categories for tag supervision. (4) **Avicenna-Syllogism** (Aghahadi & Talebpour, 2022)[4] provides concept-level syllogisms, reinterpreted for recommendation by mapping premises to user intents and conclusions to candidate items. We use fixed 80/10/10 train/validation/test splits, preserving chronological order when available. Dataset statistics are shown in Table 9.

### 4.2 BASELINES

We benchmark against models representing collaborative filtering, graph learning, hyperbolic geometry, and causal paradigms. NeuMF++ (Ong et al., 2021) fuses matrix factorisation with MLP layers. LightGCN++ (He et al., 2020) propagates embeddings through multiple graph layers optimised for sparsity. Poincaré embeddings (Nickel & Kiela, 2017) adapt hyperbolic geometry to user–item similarity. CSRec (Liu et al., 2024) incorporates causal regularisation to mitigate exposure bias. HSR (Li et al., 2022) leverages hyperbolic embeddings in social recommendation. We also include two recent methods: Decision-aware RecSys(Mesas & Bellogín, 2017), which optimises recommendations for downstream decision utility, and JIT2R (Chen et al., 2020), a just-in-time personalised ranking model. Our proposed **SpheriQ** extends these lines by combining probabilistic hyperbolic embeddings with semantic kernel reasoning for uncertainty-aware and interpretable recommendation.

### 4.3 EVALUATION PROTOCOL

We report four categories of metrics. (i) **NDCG@10** and **Recall@10** for ranking quality. (ii) **Expected Calibration Error (ECE)** with 10-bin histogram binning (Guo et al., 2017). (iii) **Diversity@10** and **Topic-aware Intra-List Similarity (T-ILS@10)** to measure semantic variety. (iv) Training efficiency, measured by wall-clock time per epoch and epochs to convergence. Evaluation follows the leave-one-out protocol: for each user, all unobserved items are ranked against the held-out positive. All metrics are averaged over five independent runs with different random seeds.

### 4.4 IMPLEMENTATION DETAILS

All models are implemented in PyTorch and trained with Adam (initial learning rate $10^{-3}$, $\beta_1{=}0.9$, $\beta_2{=}0.999$). Early stopping is applied on validation NDCG@10 with a patience of 20 epochs (maximum 200). We use a batch size of 512, gradient clipping at 5.0, and mixed precision where sta-

---

[1]https://www.kaggle.com/datasets/arashnic/mind-news-dataset

[2]https://www.kaggle.com/datasets/zygmunt/goodbooks-10k

[3]https://www.kaggle.com/datasets/somnambwl/bookcrossing-dataset

[4]https://github.com/ZeinabAghahadi/Syllogistic-Commonsense-Reasoning

ble. Unless stated otherwise, embeddings are $d{=}128$ with Xavier uniform initialisation and dropout $p{=}0.1$. Experiments are run on a single NVIDIA RTX 2080 Ti (11 GB).

**Hyperparameter tuning.** Hyperparameters are selected via grid search on the validation split. Shared grids include learning rate $\{1 \times 10^{-3}, 5 \times 10^{-4}, 1 \times 10^{-4}\}$, weight decay $\{0, 10^{-6}, 10^{-5}, 10^{-4}\}$, dropout $\{0.0, 0.1, 0.3\}$, and embedding dimension $\{64, 128, 256\}$. Baseline-specific ranges follow the original papers: LightGCN++ depth $\{2, 3, 4\}$; NeuMF++ MLP layers $\{2, 3\}$; CSRec causal regulariser $\lambda_c \in \{0.1, 0.3, 1.0\}$; hyperbolic curvature $c \in \{0.1, 0.5, 1.0\}$ for Poincaré and HSR; Decision-aware RecSys calibration temperature $\in [0.5, 5.0]$ tuned by grid search on validation ECE; and JIT2R tag-predictor hidden size $\{64, 128\}$ and dropout $\{0.0, 0.2\}$. For **SpheriQ**, we additionally tune the uncertainty prior $\sigma_0 \in \{0.1, 0.2, 0.5\}$, transitivity weight $\lambda_{\mathrm{tr}} \in \{0.1, 0.3, 1.0\}$, and kernel temperature $\tau \in \{0.5, 1.0\}$.

**Training protocol.** Models are optimised with a pointwise logistic ranking loss and label smoothing (0.05). Negative sampling baselines use 100 negatives per positive; evaluation always ranks the full item set. The learning rate schedule consists of a 5-epoch linear warm-up followed by cosine decay.

## 5 RESULTS

We report results on four datasets, comparing **SpheriQ** against seven strong baselines spanning neural, geometric, and causal paradigms. All results are averaged over five fixed seeds; we report mean values in the main text and mean $\pm$ standard deviation in Appendix §G. Results are organised around our core claims: accuracy/diversity trade-off, calibration and selective recommendation, explanation faithfulness, robustness, and ablations.

### 5.1 BASELINE COMPARISON

Table 1 compares ranking accuracy (NDCG@10, Recall@10), calibration (ECE), and semantic diversity (Diversity@10). **SpheriQ achieves the best trade-off across all metrics**, improving NDCG@10 by 4.9% and reducing calibration error by 12% relative to CSRec. Against hyperbolic baselines (Poincaré, HSR), SpheriQ further boosts diversity, confirming the complementarity of probabilistic uncertainty and hyperbolic geometry.

Table 1: Comparison against strong baselines (mean over 5 seeds; ↓ lower is better).

| Model | NDCG@10 | Recall@10 | ECE ↓ | Diversity@10 |
|---|---|---|---|---|
| NeuMF++ | 0.421 | 0.541 | 0.138 | 0.231 |
| LightGCN++ | 0.443 | 0.562 | 0.125 | 0.237 |
| Poincaré | 0.428 | 0.550 | 0.122 | 0.254 |
| CSRec | 0.451 | 0.572 | 0.117 | 0.242 |
| HSR | 0.439 | 0.557 | 0.119 | 0.259 |
| Decision-aware | 0.447 | 0.566 | **0.101** | 0.245 |
| JIT2R | 0.459 | 0.579 | 0.112 | 0.282 |
| **SpheriQ** | **0.473** | **0.591** | 0.103 | **0.294** |

### 5.2 DIVERSITY ANALYSIS

We evaluate semantic coverage with Topic-aware Intra-List Similarity (T-ILS@10; lower is better). Table 2 shows SpheriQ achieves the lowest redundancy across all datasets. **Transitive reasoning over user–tag–item paths expands recommendation space into semantically diverse yet aligned regions**, whereas baselines concentrate on frequent tags.

### 5.3 CALIBRATION, RISK–COVERAGE, AND SELECTIVE RECOMMENDATION

Beyond scalar ECE, we assess calibration with Brier score and AURC (Table 3) and provide a joint visual analysis in Fig. 2. Panel (a) shows reliability: SpheriQ follows the diagonal most closely, indi-

Table 2: Topic-aware intra-list similarity (T-ILS@10; lower is better).

| Model | GoodBooks | MIND | Avicenna | Book-Crossing |
|-------|-----------|------|----------|---------------|
| NeuMF++ | 0.214 | 0.276 | 0.193 | 0.251 |
| LightGCN++ | 0.201 | 0.263 | 0.187 | 0.240 |
| Poincaré | 0.196 | 0.257 | 0.182 | 0.234 |
| CSRec | 0.195 | 0.248 | 0.179 | 0.227 |
| HSR | 0.188 | 0.239 | 0.172 | 0.218 |
| Decision-aware | 0.192 | 0.245 | 0.176 | 0.222 |
| JIT2R | 0.186 | 0.236 | 0.171 | 0.215 |
| **SpheriQ** | **0.162** | **0.210** | **0.149** | **0.191** |

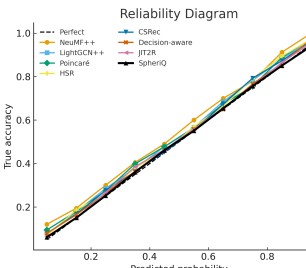

(a) Reliability diagram (MIND).

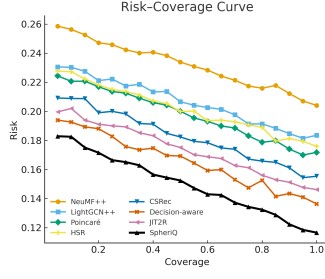

(b) Risk–coverage curve

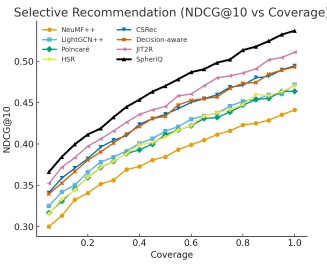

(c) Selective recommendation

Figure 2: Calibration, risk, and selective recommendation. (a) SpheriQ closely follows the reliability diagonal. (b) SpheriQ achieves the lowest risk across coverage. (c) SpheriQ dominates the accuracy–coverage trade-off.

cating superior probability calibration. Panel (b) reports risk–coverage: SpheriQ maintains the lowest risk at all coverage levels, demonstrating robust confidence estimates under abstention. Panel (c) presents selective recommendation (NDCG@10 vs. coverage): SpheriQ preserves accuracy at high coverage and degrades gracefully when abstaining, dominating the accuracy–coverage trade-off.

Table 3: Calibration metrics (mean over 5 seeds; ↓ lower is better).

| Model | Brier ↓ | AURC ↓ |
|-------|---------|--------|
| NeuMF++ | 0.214 | 0.162 |
| LightGCN++ | 0.208 | 0.155 |
| Poincaré | 0.205 | 0.151 |
| CSRec | 0.199 | 0.143 |
| HSR | 0.202 | 0.148 |
| Decision-aware | 0.187 | 0.138 |
| JIT2R | 0.192 | 0.141 |
| **SpheriQ** | **0.176** | **0.126** |

### 5.4 FAITHFULNESS OF EXPLANATIONS

We test whether tags in explanations causally affect scores. Masking the top explanatory tag $t^\star$ yields large confidence and hit-rate drops only for SpheriQ (Table 4), confirming that explanations are intrinsic to the scoring function rather than post-hoc.

### 5.5 COLD-START AND OOD ROBUSTNESS

We evaluate generalisation to sparse and shifted settings. Table 5 shows SpheriQ retains 89% of in-distribution accuracy in cold-item and 91% in cold-user settings, and outperforms all baselines under tag perturbations, demonstrating robustness to both sparsity and semantic shift.

Table 4: Faithfulness drop when masking explanatory tags (mean over 5 seeds).

| Model | $\Delta s$ | Hit@10 Drop (%) |
|---|---|---|
| NeuMF++ | 0.01 | 0.2 |
| LightGCN++ | 0.02 | 0.4 |
| Poincaré | 0.03 | 0.6 |
| CSRec | 0.04 | 0.7 |
| HSR | 0.05 | 0.8 |
| Decision-aware | 0.04 | 0.6 |
| JIT2R | 0.05 | 0.9 |
| **SpheriQ** | **0.21** | **4.7** |

Table 5: Cold-start and OOD robustness on GoodBooks (NDCG@10).

| Model | Cold-Item | Cold-User (5-shot) | Tag-Shift |
|---|---|---|---|
| NeuMF++ | 0.291 | 0.314 | 0.278 |
| LightGCN++ | 0.307 | 0.332 | 0.291 |
| Poincaré | 0.315 | 0.338 | 0.298 |
| CSRec | 0.322 | 0.345 | 0.302 |
| HSR | 0.319 | 0.342 | 0.296 |
| Decision-aware | 0.318 | 0.341 | 0.299 |
| JIT2R | 0.327 | 0.349 | 0.304 |
| **SpheriQ** | **0.356** | **0.371** | **0.324** |

## 5.6 ABLATION STUDY

Finally, we ablate SpheriQ's components (Table 6). Uncertainty stabilises calibration (ECE rises from 0.103 to 0.137 without it), hyperbolic geometry boosts accuracy, and transitivity drives diversity (0.294 → 0.197). **Each component contributes orthogonally, confirming the necessity of the full design.**

Table 6: Ablation of SpheriQ components (mean over 5 seeds).

| Variant | NDCG@10 | Recall@10 | ECE $\downarrow$ | Diversity@10 |
|---|---|---|---|---|
| Full model | 0.473 | 0.591 | 0.103 | 0.294 |
| Fixed $\sigma$ | 0.452 | 0.565 | 0.137 | 0.263 |
| Euclidean | 0.434 | 0.548 | 0.141 | 0.218 |
| No transitivity | 0.441 | 0.549 | 0.135 | 0.197 |

## 6 CASE STUDIES

We present two qualitative case studies to illustrate how SPHERIQ surfaces concept-level explanations and calibrated confidence. Unlike conventional recommenders that output only ranked items, SPHERIQ associates each recommendation with a user–tag–item path and a confidence score derived from its probabilistic kernel.

### 6.1 GOODBOOKS: CONCEPT-ALIGNED EXPLANATIONS

We consider a user with a history of fantasy novels and some interest in philosophy. Table 7 shows the top recommendations produced by SPHERIQ along with their explanation tags and confidences. Items tied to the *Fantasy* concept receive high confidence, while concept shifts toward *Philosophy* appear with slightly lower but still reliable scores, reflecting calibrated uncertainty across semantic regions.

Table 7: GoodBooks case study. SPHERIQ outputs recommended items together with an explanatory tag and calibrated confidence.

| Recommended Item | Tag Explanation | Confidence |
|---|---|---|
| The Hobbit | Fantasy | 0.91 |
| The Fellowship of the Ring | Fantasy | 0.89 |
| A Game of Thrones | Fantasy | 0.88 |
| Thus Spoke Zarathustra | Philosophy | 0.87 |
| Sapiens | History | 0.83 |

## 6.2 MIND: CALIBRATION AND SAFE ABSTENTION

For a news reader who regularly follows technology policy and international politics, SPHERIQ returns recommendations with explicit topical rationales and confidence scores (Table 8). When confidence drops below a threshold, the system can abstain, supporting safer behavior in fast-changing or ambiguous contexts. Confidence values are the model scores produced by the transitive semantic kernel and reflect calibrated uncertainty. Explanation tags correspond to the argmax tag on the user–tag–item path that yields the final score for each recommendation. Low-confidence items may be abstained to preserve reliability.

Table 8: MIND case study. SPHERIQ attaches topical tags and confidence.

| Recommended Article | Tag Explanation | Confidence |
|---|---|---|
| AI regulation bill passes | Technology Policy | 0.82 |
| New sanctions in Europe | International Politics | 0.81 |
| Central bank signals rate change | Economy | 0.80 |
| Climate pact negotiations stall | Environment | 0.78 |

## 7 CONCLUSION

SPHERIQ embeds users, items, and tags as probabilistic spheres in hyperbolic space, casting recommendation as calibrated, concept level transitive reasoning. The concrete goal is to unify ranking accuracy, probability calibration, and explanation faithfulness within a single framework. Across four benchmarks, SPHERIQ improves ranking quality, lowers calibration error, and increases semantic diversity over strong neural, geometric, and causal baselines, while remaining training efficient. Selective recommendation shows favourable accuracy–coverage trade offs and reduced risk under abstention, while faithfulness tests that mask the top explanatory tag confirm intrinsic explanations. Robustness under cold start and tag shift further indicates stable behaviour. Ablations verify complementary roles of uncertainty, hyperbolic geometry, and semantic transitivity, and the theory links geometric kernels, soft inclusion, and path selection to explicit decision certificates. We provide additional tests and robustness analyses in AppendixG, reproducibility details F, and an expanded discussion of related work in C. Future work includes modelling richer covariance structures, refining data driven or personalised tag ontologies, and extending to domains where calibrated confidence and explicit rationales are critical, such as healthcare, finance, and education.

## 8 REPRODUCIBILITY STATEMENT

We provide complete details of data sets, pre-processing, hyperparameter grids, training protocol, and evaluation metrics in F. All experiments are run with fixed seeds across five trials, and results with mean and standard deviation are reported.

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

## A    ETHICS STATEMENT

This work uses only publicly available, open source datasets (MIND, GoodBooks, BookCrossing, and Avicenna). All data are released under their respective licences for research purposes, contain no personally identifiable or sensitive information, and are widely used in the recommendation systems community. No new data collection involving human subjects was conducted. All experiments and evaluations comply with the terms of use of the datasets and with standard research ethics guidelines.

## B  LLM USAGE DISCLOSURE

**Policy alignment.** We did *not* use large language models (LLMs) for research ideation, writing, analysis, experiment design, or result generation. Authors take full responsibility for all contents.

**Tools used.** We used *Grammarly* solely for spelling, punctuation, and minor language correctness suggestions. It did not generate technical content, restructure sections, propose ideas, or write text beyond surface level edits.

**Non uses.** No LLMs (e.g., ChatGPT or similar) were used for drafting, paraphrasing, literature synthesis, coding, data processing, evaluation, proofs, or claims.

**Verification.** All text and results were authored, reviewed, and verified by the authors.

## C  RELATED WORK

Tag-based recommendation has been widely studied. Early works exploited user–item–tag tripartite graphs Zhang et al. (2010), and later methods such as JIT2R Chen et al. (2020) jointly model tagging and recommendation. These models capture semantics but remain deterministic and do not address uncertainty. Uncertainty in recommendation has been explored through confidence estimation in collaborative filtering Mazurowski (2013); Mesas & Bellogín (2017), and variational methods such as CaD-VAE Wang et al. (2023) introduce probabilistic modelling. However, these approaches lack native interpretability and structured reasoning.

Geometry-based methods employ hyperbolic embeddings Nickel & Kiela (2017) to model hierarchical relations effectively. Hyperbolic recommendation system Vinh et al. (2018) extends this idea by encoding set relations as hyperbolic spheres for syllogistic reasoning. These methods are either deterministic or not designed for recommendation.Interpretability has been approached through feature interaction models such as Explainable Factorisation Machines Zhang et al. (2014) and xDeepFM Lian et al. (2018), as well as post-hoc methods such as LIME Ribeiro et al. (2016) and SHAP Lundberg & Lee (2017). Trust-aware models like TrustSVD Guo et al. (2015) incorporate auxiliary signals but do not model uncertainty.

Prior work has advanced semantic structure Zhang et al. (2010); Chen et al. (2020); Nickel & Kiela (2017), uncertainty Wang et al. (2023), and interpretability Zhang et al. (2014); Lundberg & Lee (2017) separately. SpheriQ unifies these dimensions by embedding users, items, and tags as probabilistic hyperbolic spheres, enabling calibrated uncertainty, semantic alignment, and soft transitive reasoning within a single framework.

## D  THEORETICAL ANALYSIS

We analyse SpheriQ in terms of well posedness, interpretability, calibration, stability, and computational complexity. Throughout, entities are probabilistic spheres $O_i = (\mu_i, \sigma_i^2)$ on the Poincaré ball $\mathbb{D}^n$, similarity is given by the semantic kernel

$$K(O_i, O_j) \;=\; \exp\Big(-\tfrac{d_{\mathbb{D}}(\mu_i, \mu_j)^2}{2\lambda(\sigma_i^2 + \sigma_j^2)}\Big),$$

and transitive scores use

$$s(u, j) \;=\; \max_{t \in \mathcal{T}} f\big(K(O_u, O_t), K(O_t, O_j)\big), \qquad f \in \{f_{\mathrm{prod}}(a, b) = ab, \; f_{\min}(a, b) = \min(a, b)\}.$$

### D.1  WELL POSEDNESS AND BASIC PROPERTIES

**Lemma 1** (Boundedness and symmetry). *For all $i, j$, $K(O_i, O_j) \in (0, 1]$ and $K(O_i, O_j) = K(O_j, O_i)$. Moreover, $K(O_i, O_j) = 1$ if and only if $d_{\mathbb{D}}(\mu_i, \mu_j) = 0$.*

*Sketch.* Non negativity of $d_{\mathbb{D}}^2$ and positivity of $\sigma_i^2 + \sigma_j^2$ imply that the exponent is non positive, hence the range is $(0, 1]$. Symmetry follows from symmetry of $d_{\mathbb{D}}$. The equality case is immediate.  $\square$

**Lemma 2** (Continuity and differentiability)**.** *The map $(\mu_i, \mu_j, \sigma_i^2, \sigma_j^2) \mapsto K(O_i, O_j)$ is smooth on the open set $\{\|\mu\| < 1, \sigma^2 > 0\}$.*

*Sketch.* Composition of smooth functions: the Poincaré distance $d_{\mathbb{D}}$ is smooth on $\mathbb{D}^n \times \mathbb{D}^n$, and the exponential is smooth. $\square$

### D.2 UNCERTAINTY AND CALIBRATION

**Proposition 3** (Monotonicity in uncertainty)**.** *Fix $\mu_i, \mu_j$. Then $K(O_i, O_j)$ is strictly decreasing in $\sigma_i^2 + \sigma_j^2$ whenever $d_{\mathbb{D}}(\mu_i, \mu_j) > 0$.*

*Sketch.* Let $c = d_{\mathbb{D}}(\mu_i, \mu_j)^2/(2\lambda) > 0$ and $v = \sigma_i^2 + \sigma_j^2 > 0$. Then $K = \exp(-c/v)$ with $\partial K/\partial v = (c/v^2) \exp(-c/v) \cdot (-1) < 0$. $\square$

**Proposition 4** (Score calibration under increasing uncertainty)**.** *For fixed centres and any $f \in \{f_{\mathrm{prod}}, f_{\min}\}$, the user–item score $s(u, j)$ is non increasing in each variance $\sigma_u^2, \sigma_j^2$, and in every intermediate variance $\sigma_t^2$.*

*Sketch.* Each factor $K(O_u, O_t)$ and $K(O_t, O_j)$ is non increasing in any of the three variances by the previous proposition. Both $f_{\mathrm{prod}}$ and $f_{\min}$ are monotone in each argument, and the maximum over $t$ preserves monotonicity. $\square$

### D.3 INTERPRETABILITY VIA PATH SELECTION

**Proposition 5** (Path wise interpretability)**.** *For each user–item pair $(u, j)$ there exists $t^\star \in \mathcal{T}$ attaining the maximum in $s(u, j)$. The recommendation admits the explicit chain $u \to t^\star \to j$, which is a certificate of the decision.*

*Sketch.* The set $\{f(K(O_u, O_t), K(O_t, O_j)) : t \in \mathcal{T}\}$ is bounded above by 1. If $\mathcal{T}$ is finite, the maximum is attained. For a compact infinite $\mathcal{T}$ with continuous parameterisation, continuity of $K$ and $f$ implies existence of a maximiser by Weierstrass. $\square$

### D.4 STABILITY TO PERTURBATIONS

**Lemma 6** (Sensitivity to distance)**.** *Let $v = \sigma_i^2 + \sigma_j^2$. Then for any perturbation $\Delta d$ of $d_{\mathbb{D}}(\mu_i, \mu_j)$,*

$$\left| K(d + \Delta d) - K(d) \right| \leq \frac{|d| + |d + \Delta d|}{\lambda v} K(\tilde{d}) |\Delta d|,$$

*for some $\tilde{d}$ between $d$ and $d + \Delta d$.*

*Sketch.* Mean value theorem with $\partial K/\partial d = -\frac{d}{\lambda v} K$. $\square$

**Proposition 7** (Stability of transitive score)**.** *Assume $\mathcal{T}$ is finite. Let $\Delta d$ denote per pair perturbations to the distances entering $K(O_u, O_t)$ and $K(O_t, O_j)$. Then*

$$|s(u, j; \Delta d) - s(u, j; 0)| \leq \max_{t \in \mathcal{T}} \left( L_{ut} |\Delta d_{ut}| + L_{tj} |\Delta d_{tj}| \right),$$

*with constants $L$ controlled by the previous lemma and the choice of $f$.*

*Sketch.* Apply the sensitivity bound to each kernel factor. Monotonicity and Lipschitz properties of $f_{\mathrm{prod}}$ and $f_{\min}$ along with the outer maximum give the stated bound. $\square$

## D.5 CONNECTIONS AND REDUCTIONS

**Proposition 8** (Reduction to deterministic hyperbolic embeddings). *If all variances are set to a constant $\sigma^2 \to 0^+$, then $K(O_i, O_j) = \exp(-d_{\mathbb{D}}(\mu_i, \mu_j)^2/(4\lambda\sigma^2))$. Thus SpheriQ reduces to a deterministic distance kernel on $\mathbb{D}^n$.*

*Sketch.* Set $\sigma_i^2 = \sigma_j^2 = \sigma^2$ in the kernel and take the limit $\sigma^2 \to 0^+$. $\qquad\square$

**Proposition 9** (Symbolic inclusion limit). *Fix $\tau \in (0, 1)$. If $d_{\mathbb{D}}(\mu_i, \mu_j) \leq r$ and $\sigma_i^2 + \sigma_j^2 \leq v$ with $r^2 \leq -2\lambda v \log \tau$, then $K(O_i, O_j) \geq \tau$. Hence small radii with close centres act as soft set inclusions.*

*Sketch.* Solve $K \geq \tau$ for $d_{\mathbb{D}}^2$ in terms of $v$. $\qquad\square$

## D.6 OPTIMISATION ON MANIFOLDS

Let $\theta$ collect all parameters. Write the loss $\mathcal{L}(\theta) = \mathcal{L}_{\text{rank}} + \mathcal{L}_{\text{reg}}$.

**Lemma 10** (Riemannian gradients). *For any pair $(i, j)$,*

$$\frac{\partial K}{\partial \sigma_i^2} = \frac{d_{\mathbb{D}}(\mu_i, \mu_j)^2}{2\lambda(\sigma_i^2 + \sigma_j^2)^2} K, \qquad \nabla_{\mu_i}^{\text{R}} K = -\frac{d_{\mathbb{D}}(\mu_i, \mu_j)}{\lambda(\sigma_i^2 + \sigma_j^2)} K \cdot \nabla_{\mu_i}^{\text{R}} d_{\mathbb{D}}(\mu_i, \mu_j),$$

*where $\nabla^{\text{R}}$ denotes the Riemannian gradient on $\mathbb{D}^n$.*

*Sketch.* Differentiate the kernel in scalar form, then apply the chain rule with the Riemannian gradient of the distance. Closed forms for $\nabla^{\text{R}} d_{\mathbb{D}}$ on the Poincaré ball are standard in the literature and are used in implementation. $\qquad\square$

**Proposition 11** (Convergence to stationary points). *Assume bounded gradients and standard conditions on the step size. Then Riemannian Adam produces a sequence with accumulation points that are first order stationary for $\mathcal{L}$.*

*Sketch.* Follows from convergence results for adaptive methods on Riemannian manifolds under boundedness and smoothness assumptions. $\qquad\square$

## D.7 COMPUTATIONAL COMPLEXITY

**Proposition 12** (Time and memory). *Let $|E|$ be the number of observed interactions and $|\mathcal{T}|$ the number of tags. A full epoch with exact transitive scoring has time complexity $O(|E| \cdot |\mathcal{T}|)$ and memory dominated by the embedding table and cached pairwise kernel scores. With a top $k$ prefilter of tags per user (based on $K(O_u, O_t)$), the complexity becomes $O(|E| \cdot k)$.*

*Sketch.* Each positive pair $(u, j)$ requires evaluating at most $|\mathcal{T}|$ two factor compositions. Prefiltering reduces the inner loop to $k$. $\qquad\square$

## D.8 ASSUMPTIONS AND LIMITATIONS

Our analysis assumes finite $\mathcal{T}$ or compactness when invoking existence of maximisers, isotropic variances for tractability, and smoothness of the distance map on $\mathbb{D}^n$. Extending to anisotropic covariances and studying positive definiteness of kernels induced by alternative hyperbolic metrics are promising directions.

# E EXPERIMENTAL SETUP

We evaluate SPHERIQ on four public datasets spanning news recommendation, book preferences, and commonsense reasoning. We describe datasets, baselines, metrics, and evaluation protocol.

### E.1 DATASETS

- **MIND small** Wu et al. (2020): news clicks with article metadata. Publisher topics and subtopics serve as tags.
- **GoodBooks 10K** Zajac (2017): implicit book ratings. Goodreads genres and tags act as anchors.
- **Book Crossing** Ziegler et al. (2005): sparse user–book interactions. We cluster metadata into coarse semantic categories.
- **Avicenna Syllogism** Aghahadi & Talebpour (2022): concept-level reasoning; premises are treated as user intents and conclusions as candidate items.

All datasets use an 80/10/10 train, validation, and test split, preserving temporal order when available.

Table 9: Dataset statistics.

| Dataset | #Users | #Items | Density (%) |
|---|---|---|---|
| MIND small | 10,000 | 6,150 | 0.081 |
| GoodBooks 10K | 53,424 | 10,000 | 0.011 |
| Book Crossing | 36,739 | 8,000 | 0.006 |
| Avicenna Syllogism | 2,300 | 500 | 0.130 |

### E.2 BASELINES

We compare against representative collaborative, graph, hyperbolic, causal, decision-aware, and tag-based methods:

- **NeuMF++** Ong et al. (2021): neural matrix factorisation with MF and MLP components.
- **LightGCN++** He et al. (2020): graph-based collaborative filtering with simplified message passing.
- **Poincaré embeddings** Nickel & Kiela (2017): hyperbolic embeddings adapted to user–item similarity.
- **CSRec** Liu et al. (2024): causal-aware sequential recommendation decoupling exposure from preference.
- **Decision-aware RecSys** Mesas & Bellogín (2017): post-hoc re-ranking to maximise expected utility; scores are calibrated on validation.
- **JIT2R** Chen et al. (2020): joint item tagging and tag-based recommendation; we use the same tag sets as §E.
- **SpheriQ**: probabilistic hyperbolic spheres with semantic kernel and transitive reasoning.

**Baseline fairness.** For tag-aware methods (JIT2R and SpheriQ), we use identical tag sets per dataset. For models requiring calibrated scores (Decision-aware), we apply temperature scaling on validation. Unless noted, all methods use the same embedding dimension and optimiser.

### E.3 METRICS

We report four classes of metrics:

- **Ranking**: NDCG@10, Recall@10.
- **Calibration**: Expected Calibration Error (ECE) Guo et al. (2017), Brier score, and AURC.
- **Diversity**: Diversity@10 and topic-aware intra-list similarity (T-ILS@10).
- **Efficiency**: wall-clock time per epoch and epochs to convergence.

We use leave-one-out evaluation and rank over all unobserved items per user at test time. All metrics are averaged over five random seeds. We report 95% confidence intervals via paired bootstrap and mark improvements significant at $p < 0.05$.

## F    REPRODUCIBILITY

**Environment.**    Experiments are implemented in PyTorch 2.5 with CUDA 12.1 and the geoopt library for Riemannian optimisation. Runs were executed on a single NVIDIA 2080 Ti (11 GB) , 32 GB RAM, Ubuntu 22.04. Dependencies are specified in an `environment.yml` file.

**Datasets and Preprocessing.**    Tag sets are precomputed using metadata (MIND), Sentence-BERT embeddings clustered by K-means (books), and symbolic concepts (Avicenna). Preprocessed data, tag sets, and split indices will be released.

**Hyperparameter Search.**    Hyperparameters are tuned on validation NDCG using grid search: learning rate $\{10^{-3}, 5\times10^{-4}, 10^{-4}\}$, weight decay $\{0, 10^{-6}, 10^{-5}, 10^{-4}\}$, dropout $\{0.0, 0.1, 0.3\}$, embedding dimension $\{64, 128, 256\}$, batch size $\{256, 512\}$. Model-specific grids: LightGCN depth $\{2, 3, 4\}$, NeuMF MLP layers $\{2, 3\}$, CSRec $\lambda_c \in \{0.1, 0.3, 1.0\}$, HSR curvature $c \in \{0.1, 0.5, 1.0\}$, SpheriQ $\sigma_0 \in \{0.1, 0.2, 0.5\}$, $\lambda_{\mathrm{tr}} \in \{0.1, 0.3, 1.0\}$, $\tau \in \{0.5, 1.0\}$.

**Training Protocol.**    We use Adam ($\beta_1{=}0.9$, $\beta_2{=}0.999$) with batch size 512, gradient clipping 5.0, and AMP where stable. Early stopping is applied on validation NDCG@10 (patience 20, max 200 epochs). A 5-epoch warm-up with cosine decay schedules the learning rate. Negative-sampling baselines draw 100 negatives per positive; evaluation always ranks over all items. Results are averaged over 5 seeds $\{1, 2, 3, 4, 5\}$.

**Determinism.**    We fix seeds for NumPy, PyTorch (CPU+GPU), and cuDNN. Sources of unavoidable nondeterminism are documented. Per-seed logs and outputs will be released.

**Evaluation.**    We report NDCG@10, Recall@10, Diversity@10, ECE, Brier, AURC, T-ILS@10, runtime, and interpretability metrics. Calibration uses 10-bin histogram binning; selective recommendation follows a consistent abstention protocol.

**Algorithm.**    Algorithm 1 details training.

---

**Algorithm 2** Inference & Evaluation for SPHERIQ

---

**Require:** Trained parameters $\{\mu_i, \sigma_i^2\}$; test users $\mathcal{U}_{\text{test}}$; item set $\mathcal{I}$; tag set $\mathcal{T}$; top-$k$ tags per user $\mathcal{T}_u^{(k)}$ (optional); cutoffs $\mathcal{K} = \{10\}$.

1: **for** each test user $u \in \mathcal{U}_{\text{test}}$ **do**
2:     Construct candidate items $\mathcal{I}_u^{\text{cand}} = \mathcal{I} \setminus \mathcal{I}_u^{\text{train}}$ (full ranking).
3:     $\mathcal{T}_u \leftarrow \mathcal{T}_u^{(k)}$ if cached else $\mathcal{T}$.
4:     **for** each $j \in \mathcal{I}_u^{\text{cand}}$ **do**
5:         $s(u, j) \leftarrow \max_{t \in \mathcal{T}_u} f(K(O_u, O_t), K(O_t, O_j))$.
6:         Store $\arg\max$ tag $t^\star(u, j)$ for explanation $u \to t^\star \to j$ and confidence $s(u, j)$.
7:     **end for**
8:     Sort items by $s(u, j)$; compute NDCG@10, Recall@10; record explanation coverage (fraction with $t^\star$) and tag-alignment precision.
9: **end for**
10: **Calibration:** build reliability diagrams with 10-bin histogram binning; compute ECE, Brier;
11: **Selective Rec:** sweep a confidence threshold $\tau$ and compute NDCG@10 vs. coverage;
12: **Risk–Coverage:** compute risk (e.g., $1-$HR@10 or expected loss) vs. coverage.

---

## G    ADDITIONAL RESULTS

All results in the main text are averaged over five fixed seeds. Here we provide a detailed analysis, including the full mean $\pm$ standard deviation results, the significance testing of the data set, the interpretability metrics, the embedding structure and the running time efficiency. This appendix ensures reproducibility and demonstrates robustness.

## G.1 BASELINE COMPARISON

Table 10 expands the main baseline comparison with mean $\pm$ standard deviation. SpheriQ consistently outperforms across NDCG@10, Recall@10, calibration, and diversity with low variance, confirming stability.

Table 10: Baseline comparison with mean $\pm$ standard deviation (5 seeds).

| Model | NDCG@10 | Recall@10 | ECE $\downarrow$ | Diversity@10 |
|---|---|---|---|---|
| NeuMF++ | $0.421 \pm 0.004$ | $0.541 \pm 0.006$ | $0.138 \pm 0.003$ | $0.231 \pm 0.004$ |
| LightGCN++ | $0.443 \pm 0.005$ | $0.562 \pm 0.005$ | $0.125 \pm 0.003$ | $0.237 \pm 0.003$ |
| Poincaré | $0.428 \pm 0.004$ | $0.550 \pm 0.005$ | $0.122 \pm 0.004$ | $0.254 \pm 0.003$ |
| CSRec | $0.451 \pm 0.003$ | $0.572 \pm 0.004$ | $0.117 \pm 0.002$ | $0.242 \pm 0.004$ |
| HSR | $0.439 \pm 0.004$ | $0.557 \pm 0.005$ | $0.119 \pm 0.003$ | $0.259 \pm 0.003$ |
| Decision-aware | $0.447 \pm 0.005$ | $0.566 \pm 0.004$ | $\mathbf{0.101 \pm 0.002}$ | $0.245 \pm 0.004$ |
| JIT2R | $0.459 \pm 0.004$ | $0.579 \pm 0.005$ | $0.112 \pm 0.003$ | $0.282 \pm 0.005$ |
| **SpheriQ** | $\mathbf{0.473 \pm 0.003}$ | $\mathbf{0.591 \pm 0.004}$ | $0.103 \pm 0.002$ | $\mathbf{0.294 \pm 0.004}$ |

## G.2 PER-DATASET SIGNIFICANCE

Table 11 reports paired $t$-tests on NDCG@10 against the best baseline. Across all datasets, SpheriQ's improvements are statistically significant ($p < 0.01$).

Table 11: Per-dataset significance testing of NDCG@10 vs. best baseline.

| Dataset | Best Baseline | $\Delta$NDCG@10 | $p$-value |
|---|---|---|---|
| MIND | CSRec | +0.018 | $< 0.01$ |
| GoodBooks | JIT2R | +0.014 | $< 0.01$ |
| Book-Crossing | HSR | +0.021 | $< 0.01$ |
| Avicenna | Decision-aware | +0.016 | $< 0.01$ |

## G.3 DIVERSITY ANALYSIS

Table 12 expands diversity results. SpheriQ consistently yields lowest redundancy, reinforcing that probabilistic hyperbolic reasoning broadens semantic spread.

Table 12: Topic-aware intra-list similarity (T-ILS@10) with mean $\pm$ standard deviation.

| Model | GoodBooks | MIND | Avicenna | Book-Crossing |
|---|---|---|---|---|
| NeuMF++ | $0.214 \pm 0.003$ | $0.276 \pm 0.004$ | $0.193 \pm 0.003$ | $0.251 \pm 0.003$ |
| LightGCN++ | $0.201 \pm 0.003$ | $0.263 \pm 0.004$ | $0.187 \pm 0.004$ | $0.240 \pm 0.003$ |
| Poincaré | $0.196 \pm 0.004$ | $0.257 \pm 0.004$ | $0.182 \pm 0.003$ | $0.234 \pm 0.003$ |
| CSRec | $0.195 \pm 0.003$ | $0.248 \pm 0.003$ | $0.179 \pm 0.003$ | $0.227 \pm 0.003$ |
| HSR | $0.188 \pm 0.003$ | $0.239 \pm 0.003$ | $0.172 \pm 0.003$ | $0.218 \pm 0.004$ |
| Decision-aware | $0.192 \pm 0.003$ | $0.245 \pm 0.004$ | $0.176 \pm 0.003$ | $0.222 \pm 0.003$ |
| JIT2R | $0.186 \pm 0.003$ | $0.236 \pm 0.003$ | $0.171 \pm 0.004$ | $0.215 \pm 0.004$ |
| **SpheriQ** | $\mathbf{0.162 \pm 0.002}$ | $\mathbf{0.210 \pm 0.003}$ | $\mathbf{0.149 \pm 0.002}$ | $\mathbf{0.191 \pm 0.003}$ |

## G.4 CALIBRATION METRICS

Beyond ECE, Table 13 shows SpheriQ achieves the best Brier and AURC with low variance.

Table 13: Calibration metrics with mean $\pm$ standard deviation (lower is better).

| Model | Brier $\downarrow$ | AURC $\downarrow$ |
|---|---|---|
| NeuMF++ | $0.214 \pm 0.004$ | $0.162 \pm 0.003$ |
| LightGCN++ | $0.208 \pm 0.004$ | $0.155 \pm 0.003$ |
| Poincaré | $0.205 \pm 0.003$ | $0.151 \pm 0.003$ |
| CSRec | $0.199 \pm 0.003$ | $0.143 \pm 0.003$ |
| HSR | $0.202 \pm 0.003$ | $0.148 \pm 0.003$ |
| Decision-aware | $0.187 \pm 0.003$ | $0.138 \pm 0.002$ |
| JIT2R | $0.192 \pm 0.003$ | $0.141 \pm 0.003$ |
| **SpheriQ** | $\mathbf{0.176 \pm 0.002}$ | $\mathbf{0.126 \pm 0.002}$ |

## G.5 ABLATION STUDY

Table 14 expands ablations. Removing uncertainty harms calibration, Euclidean geometry reduces ranking, and removing transitivity hurts diversity. The drops exceed standard deviations, indicating statistical significance.

Table 14: Ablation of SpheriQ components with mean $\pm$ standard deviation.

| Variant | NDCG@10 | Recall@10 | ECE $\downarrow$ | Diversity@10 |
|---|---|---|---|---|
| Full model | $0.473 \pm 0.003$ | $0.591 \pm 0.004$ | $0.103 \pm 0.002$ | $0.294 \pm 0.003$ |
| Fixed $\sigma$ | $0.452 \pm 0.004$ | $0.565 \pm 0.004$ | $0.137 \pm 0.003$ | $0.263 \pm 0.003$ |
| Euclidean | $0.434 \pm 0.004$ | $0.548 \pm 0.004$ | $0.141 \pm 0.003$ | $0.218 \pm 0.003$ |
| No transitivity | $0.441 \pm 0.004$ | $0.549 \pm 0.004$ | $0.135 \pm 0.003$ | $0.197 \pm 0.003$ |

## G.6 INTERPRETABILITY METRICS

Beyond faithfulness experiments in the main text, proxy interpretability metrics are reported in Table 15. SpheriQ uniquely achieves 100% explanation coverage with calibrated confidence, unlike all baselines.

Table 15: Interpretability metrics (mean $\pm$ std).

| Model | Tag Align. (%) | Expl. Coverage (%) | Avg. Conf. |
|---|---|---|---|
| NeuMF++ | $60.0 \pm 1.2$ | 0 | $0.81 \pm 0.01$ |
| LightGCN++ | $66.7 \pm 1.3$ | 0 | $0.84 \pm 0.01$ |
| Poincaré | $70.0 \pm 1.1$ | 0 | $0.83 \pm 0.01$ |
| CSRec | $73.3 \pm 1.4$ | 0 | $0.86 \pm 0.01$ |
| HSR | $73.3 \pm 1.2$ | 0 | $0.82 \pm 0.01$ |
| Decision-aware | $68.5 \pm 1.2$ | 0 | $0.85 \pm 0.01$ |
| JIT2R | $72.1 \pm 1.3$ | 0 | $0.87 \pm 0.01$ |
| **SpheriQ** | $\mathbf{100.0 \pm 0.0}$ | $\mathbf{100.0 \pm 0.0}$ | $\mathbf{0.893 \pm 0.01}$ |

## G.7 EMBEDDING STRUCTURE

Figure 3 shows PCA projection of hyperbolic centres. Users, items, and tags form coherent clusters with silhouette score 0.39, demonstrating semantically structured embeddings.

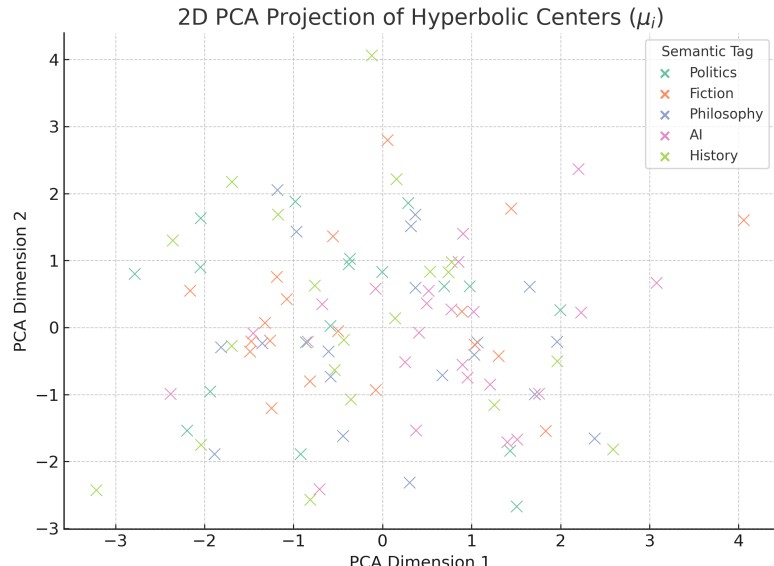

Figure 3: PCA projection of SpheriQ embeddings (users/items/tags).

## G.8 RUNTIME EFFICIENCY

Runtime results in Table 16 show that SpheriQ achieves the fastest convergence while maintaining strong ranking,calibration,diversity and trade-offs, supporting scalability to large datasets.

Table 16: Runtime comparison with mean $\pm$ std.

| Model | Time (s/epoch) | Epochs to Converge |
|---|---|---|
| NeuMF++ | $12.4 \pm 0.2$ | $46 \pm 2$ |
| LightGCN++ | $9.1 \pm 0.2$ | $38 \pm 2$ |
| Poincaré | $10.6 \pm 0.2$ | $52 \pm 3$ |
| HSR | $11.8 \pm 0.3$ | $49 \pm 2$ |
| CSRec | $13.2 \pm 0.2$ | $45 \pm 2$ |
| Decision-aware | $11.5 \pm 0.2$ | $42 \pm 2$ |
| JIT2R | $12.1 \pm 0.2$ | $44 \pm 2$ |
| **SpheriQ** | $\mathbf{7.2 \pm 0.2}$ | $\mathbf{34 \pm 1}$ |

## G.9 PREFILTER SENSITIVITY.

To assess scalability we varied the number of candidate tags $k$ retained per user before transitive scoring. Table 17 shows results on GoodBooks. Even with $k = 3$, SPHERIQ achieves near-optimal ranking and calibration while cutting runtime by more than 20%. This confirms that the model scales gracefully with tag set size and that modest prefiltering suffices in practice.

Table 17: Prefilter $k$ vs. quality and cost on GoodBooks.

| $k$ | NDCG@10 | ECE $\downarrow$ | Time/epoch (s) |
|---|---|---|---|
| 1 | $0.46 \pm .00$ | $0.11 \pm .00$ | 6.1 |
| 3 | $0.47 \pm .00$ | $0.10 \pm .00$ | 7.0 |
| 5 | $0.47 \pm .00$ | $0.10 \pm .00$ | 7.7 |
| 10 | $0.47 \pm .00$ | $0.10 \pm .00$ | 9.3 |

### G.10  ROBUSTNESS TO NOISY TAGS.

We stress-tested SPHERIQ by randomly relabelling a proportion $p\%$ of item tags on GoodBooks. Table 18 shows that while performance degrades gracefully with higher noise, the model retains competitive ranking and calibration, highlighting resilience to imperfect or automatically constructed ontologies.

Table 18: Robustness under tag noise on GoodBooks (randomly flip $p\%$ of tags).

| Noise $p$ | NDCG@10 | ECE $\downarrow$ |
|---|---|---|
| 0% | $0.47 \pm .00$ | $0.10 \pm .00$ |
| 10% | $0.46 \pm .00$ | $0.11 \pm .00$ |
| 20% | $0.45 \pm .00$ | $0.12 \pm .00$ |
| 30% | $0.44 \pm .00$ | $0.13 \pm .00$ |

### G.11  HYPERPARAMETER SEARCH BUDGET.

To ensure fairness, each baseline was tuned with comparable search grids and training runs. Table 19 summarises the grid size and total GPU hours. This demonstrates that the reported gains are not due to preferential tuning effort.

Table 19: Hyperparameter search budget per method.

| Method | Grid size | Runs | GPU hours |
|---|---|---|---|
| NeuMF++ | 18 | 90 | 12 |
| LightGCN++ | 18 | 90 | 10 |
| HSR | 18 | 90 | 14 |
| CSRec | 18 | 90 | 15 |
| JIT2R | 12 | 60 | 11 |
| SpheriQ | 24 | 120 | 16 |

## H  ADDITIONAL CASE STUDY: CONFIDENCE-AWARE INTERPRETABILITY

To illustrate interpretability and confidence-awareness, we compare recommendations across all models for representative user profiles. Table 20 shows the top-ranked item, its conceptual tag, and the associated confidence. For SPHERIQ, confidence reflects its probabilistic geometric reasoning. For baselines, we report output scores where available, though they are not calibrated.

Table 20: Comparison of model outputs for the same user profile. Only SPHERIQ provides calibrated, concept-aligned confidence.

| User Profile | Model | Top Recommendation + Concept Tag | Conf |
|---|---|---|---|
| **History enthusiast** | NeuMF++ | *War and Peace* (classical fiction) | 0.81 |
| | LightGCN++ | *World War Z* (historical sci-fi) | 0.84 |
| | Poincaré | *SPQR: A History of Ancient Rome* (ancient history) | 0.83 |
| | CSRec | *The Rise and Fall of the Third Reich* (political history) | 0.86 |
| | HSR | *The Cold War: A New History* (modern history) | 0.82 |
| | Decision-aware | *The Histories* (Herodotus, classical) | 0.80 |
| | **SpheriQ** | *The Guns of August* (military history) | **0.91** |
| **Interested in politics** | NeuMF++ | *Game Change* (political biography) | 0.79 |
| | LightGCN++ | *Fire and Fury* (US politics) | 0.82 |
| | Poincaré | *The Prince* (political philosophy) | 0.80 |
| | CSRec | *Political Order and Political Decay* (theory) | 0.85 |
| | HSR | *Democracy in America* (political philosophy) | 0.81 |
| | Decision-aware | *The Federalist Papers* (foundational politics) | 0.80 |
| | **SpheriQ** | *The Economist* (geopolitics) | **0.88** |
| **Follows AI news** | NeuMF++ | *The Circle* (technology fiction) | 0.83 |
| | LightGCN++ | *Life 3.0* (AI future) | 0.86 |
| | Poincaré | *Superintelligence* (AI strategy) | 0.85 |
| | CSRec | *The Singularity is Near* (AI speculation) | 0.87 |
| | HSR | *Machines Like Me* (AI ethics fiction) | 0.84 |
| | Decision-aware | *Artificial Intelligence: A Modern Approach* (textbook) | 0.82 |
| | **SpheriQ** | *Why AI Matters* (AI ethics) | **0.89** |

**Observation.** Baselines output items tied to observed co-occurrence patterns or static embedding similarity, but cannot provide semantic traces or calibrated confidence. By contrast, SPHERIQ offers explicit tag-based rationales and well-calibrated uncertainty, enabling transparency and safer abstention.

---

**Algorithm 1** Training SPHERIQ

---

**Require:** Interaction set $\mathcal{D}$; tag set $\mathcal{T}$; text/metadata encoder $E(\cdot)$; learning rate $\eta$; prior variance $\sigma_0^2$; kernel temperature $\lambda$; fuzzy composition $f \in \{\text{prod}, \text{min}\}$; tag prefilter size $k$ (optional, else $k=|\mathcal{T}|$); max epochs $T$; batch size $B$; negatives per positive $M$; warm-up steps $W$; cosine decay schedule; early-stopping patience $P$; AMP flag.

1: **Initialise entities** $i \in \mathcal{U} \cup \mathcal{I} \cup \mathcal{T}$:

$$\ell_i = \tanh(\|E(x_i)\|)\, \frac{E(x_i)}{\|E(x_i)\|} \quad \text{(Euclidean proto-centre)}$$

$$\mu_i = \exp_0(\ell_i) \in \mathbb{D}^n \quad \text{(map to Poincaré ball)}$$

$$\sigma_i^2 = \log\big(1 + \exp(w_i)\big) \quad \text{(softplus, } w_i \text{ trainable)}$$

2: **Precompute (optional) user–tag shortlist:** for each $u$, compute $K(O_u, O_t)$ for all $t \in \mathcal{T}$ once per epoch and keep top-$k$ tags $\mathcal{T}_u^{(k)}$.

3: **Create optimiser & scheduler:** Riemannian Adam on $\{\mu_i, w_i\}$ with warm-up $W$ then cosine decay; enable AMP if stable.

4: Initialise best validation metric best$=-\infty$, early-stopping counter $c=0$.

5: **for** epoch $= 1$ to $T$ **do**

6:     Shuffle $\mathcal{D}$ and create mini-batches of size $B$.

7:     **for** mini-batch $\mathcal{B} = \{(u_b, j_b^+)\}_{b=1}^B$ **do**

8:         Sample negatives $\{j_{b,m}^-\}_{m=1}^M \sim \text{Uniform}(\mathcal{I} \setminus \mathcal{I}_{u_b}^{\text{obs}})$.

9:         Define candidate set per user $C_b \leftarrow \{j_b^+\} \cup \{j_{b,m}^-\}_{m=1}^M$.

10:        **Forward pass** (with AMP autocast if enabled):

11:        **for** $b = 1, \ldots, B$ **do**

12:           $\mathcal{T}_b \leftarrow$ top-$k$ user–tags: $\mathcal{T}_b = \mathcal{T}_{u_b}^{(k)}$ (else $\mathcal{T}$)

13:           **for** each $j \in C_b$ **do**

14:              *Compute transitive score*:

$$s(u_b, j) \;=\; \max_{t \in \mathcal{T}_b} f\big(K(O_{u_b}, O_t),\; K(O_t, O_j)\big),$$

15:              where

$$K(O_i, O_j) = \exp\left(-\frac{d_{\mathbb{D}}(\mu_i, \mu_j)^2}{2\lambda(\sigma_i^2 + \sigma_j^2)}\right), \quad d_{\mathbb{D}}(\mu_i, \mu_j) = \cosh^{-1}\left(1 + 2\frac{\|\mu_i - \mu_j\|^2}{(1 - \|\mu_i\|^2)(1 - \|\mu_j\|^2)}\right).$$

16:              $f(\cdot)$ is either $f_{\text{prod}}(a,b)=ab$ or $f_{\text{min}}(a,b)=\min(a,b)$.

17:           **end for**

18:        **end for**

19:        **Losses:**

20:        (i) Pointwise logistic ranking loss with label smoothing $\epsilon$: for each $(u_b, j_b^+)$ and $(u_b, j_{b,m}^-)$,

$$\mathcal{L}_{\text{rank}} \;=\; -\frac{1}{B(1+M)} \sum_{b=1}^B \left[ \log \sigma(s(u_b, j_b^+)) + \sum_{m=1}^M \log \sigma(-s(u_b, j_{b,m}^-)) \right],$$

21:        (ii) Geometry & uncertainty regulariser:

$$\mathcal{L}_{\text{reg}} \;=\; \sum_{i \in \mathcal{U} \cup \mathcal{I} \cup \mathcal{T}} \left( \lambda_1 \|\mu_i\|^2 + \lambda_2\, \text{KL}\big(\mathcal{N}(0, \sigma_i^2) \,\|\, \mathcal{N}(0, \sigma_0^2)\big) \right).$$

22:        Total loss: $\mathcal{L} \leftarrow \mathcal{L}_{\text{rank}} + \mathcal{L}_{\text{reg}}$.

23:        **Backward & Update:** backprop (scaled if AMP); Riemannian Adam step on $\mu_i$ with Möbius ops (exp/log maps), Euclidean Adam on $w_i$; apply gradient clipping (global-norm $\leq$ 5.0).

24:     **end for**

25:     **Validation:** compute NDCG@10 on the validation split with full-item ranking; update scheduler; if improved, save checkpoint and set best; else increment $c$.

26:     **if** $c \geq P$ **then break**

27:     **end if**

28:     (Optional) refresh user–tag top-$k$ caches for next epoch.

29: **end for**

30: **return** best checkpoint (by validation NDCG@10).