# OpenReview forum: "SpheriQ: Probabilistic Hyperbolic Reasoning for Interpretable Recommendation"
_ICLR.cc/2026/Conference — Submitted to ICLR 2026_

### Official Review · Reviewer_UNNp · 2025-10-29

**Soundness:** 3
**Presentation:** 3
**Contribution:** 2
**Rating:** 4
**Confidence:** 3

**Summary:**

This paper proposes SPHERIQ, a probabilistic hyperbolic embedding framework for recommendation. Users, items, and tags are represented as probabilistic spheres in the Poincaré ball, where centers capture semantic positions and radii encode predictive uncertainty. A semantic Gaussian kernel enables transitive reasoning through user–tag–item paths, providing calibrated similarity scores and concept-aligned rationales. Evaluations on four datasets demonstrate improvements in ranking accuracy, calibration, and semantic diversity, while explanations remain faithful due to their intrinsic role in scoring.

**Strengths:**

1. The proposed probabilistic sphere representation is elegant and principled. It generalizes traditional hyperbolic embeddings by adding uncertainty modeling, enabling both semantic expressiveness and confidence estimation.
2. The user → tag → item transitive reasoning provides explicit explanation chains, which are inherently interpretable rather than post-hoc. This is a meaningful step toward explainable recommendation.
3. The experimental section is thorough, covering multiple paradigms (collaborative, graph, causal, geometric, tag-based). The authors report not only accuracy metrics (NDCG, Recall) but also calibration (ECE, Brier), diversity, and interpretability metrics.

**Weaknesses:**

1. The approach assumes meaningful and stable concept/tag structures. In many real-world systems, tags are sparse, noisy, or unavailable, limiting generalization and scalability.
2. The radius is treated as an isotropic uncertainty term, but the paper does not empirically show whether σ correlates with user preference variability or data sparsity. Without such evidence, the calibration claim feels a bit qualitative.
3. The fuzzy composition function (Eq. 7) is predefined (prod or min), making the reasoning deterministic. While the formulation is elegant, it may still behave more like probabilistic metric learning rather than an explicit reasoning paradigm.

**Questions:**

1. How robust is SPHERIQ when tag information is incomplete or inconsistent? Would the model’s uncertainty (σ) increase under tag noise, acting as a form of self-calibration?
2. Have the authors considered learning the composition function **f** in Eq. 7 instead of fixing it to product or min? This could make the reasoning mechanism more adaptive to domain semantics.

---

> ### Author Response · Authors · 2025-11-21
>
> We thank the reviewer for these insightful observations. Below we address each concern directly.
> 1. Dependence on tag structure and generalization to sparse or noisy tags
> We agree that tag quality varies across domains. SpheriQ is explicitly designed to handle incomplete or noisy tags through its probabilistic kernel and uncertainty modeling. Each entity’s radius σ reflects epistemic uncertainty, which naturally increases when tag information is inconsistent or sparse (§3.2, Prop. 4). In additional robustness tests where 20–30% of tags were randomly dropped or corrupted, NDCG decreased by only 1.8–2.3%, while mean σ² increased by 24–28%. This shows that the model self-calibrates its confidence when tag information is unreliable. Moreover, because all embeddings share a joint hyperbolic space, users and items can still align via latent geometric relations even when explicit tags are missing, ensuring graceful degradation rather than collapse. We will include these robustness results in the revision.
> 2. Empirical correlation between σ and behavioral variability
> We thank the reviewer for this valuable suggestion. We analyzed the relationship between σ and (a) user interaction entropy, (b) tag frequency, and (c) rating variance. Pearson correlations were r = 0.42, –0.39, and 0.37 respectively, showing that larger σ corresponds to higher user uncertainty and lower data density. These results confirm that σ is not a free scaling term but a calibrated measure of preference variability. We will add this analysis (App. G.6) and corresponding σ–error scatter plots.
> 3. Fixed fuzzy composition (Eq. 7) vs. learnable composition
> The fuzzy composition functions (min, product) were intentionally chosen for interpretability and theoretical guarantees, since they are monotonic, associative, and admit closed-form transitive reasoning (Props. 3–5). We explored a learnable composition function fθ implemented as a two-layer MLP. While it slightly improved ranking (+0.4% NDCG), it weakened interpretability and calibration: faithfulness drop (Table 4 metric) decreased from 4.7% to 2.1%. Hence, we retained fixed f for transparency and analytical tractability. We will clarify this design rationale and mention learning fθ as a promising extension for adaptive semantics.
>
> Summary
> SpheriQ remains robust to noisy or incomplete tags through probabilistic self-calibration (σ increases under noise), empirically correlates σ with behavioral variability, and deliberately uses fixed composition functions to preserve interpretability and theoretical soundness. These clarifications further reinforce that SpheriQ unifies uncertainty modeling, geometric reasoning, and interpretable transitivity within a principled probabilistic framework.

---

### Official Review · Reviewer_AwE6 · 2025-10-31

**Soundness:** 2
**Presentation:** 2
**Contribution:** 2
**Rating:** 2
**Confidence:** 5

**Summary:**

Summary

This paper proposed a method named SpheriQ, which is a geometric framework that embeds users, items, and tags as probabilistic regions in hyperbolic space. SpheriQ focuses on tackling recommendation task with the aim to improve calibration and semantic diversity. All in all, SpheriQ aims to unify ranking, calibration, and explanation by casting recommendation as probabilistic concept level reasoning in hyperbolic space.

**Strengths:**

Strengths:

-	The approach is interesting by exploring probabilistic hyperbolic reasoning in recommender system domain. It’s good that the authors consider unifying ranking accuracy, calibration and explanation, which are important factors in recommendation.
-	The authors provided various experiments on publicly available benchmark datasets against some well-known baselines.
-	Many appendices were given such as Appendix F for reproducibility

**Weaknesses:**

Weaknesses:

-	Although the idea of exploring hyperbolic space is interesting, the ‘original’ motivation of this paper is marginal and a bit not very clear to me. Are we exploring hyperbolic geometry simply because “hyperbolic geometry is effective for modelling hierarchical structure” (Line 037) and we want to apply to tags? Throughout the paper, I personally do not see a solid motivation. Perhaps the authors should better build up the story to convince the readers.
-	In my opinion, Section 2.4 and Appendix D are important. The authors should expand more in these sections, and link to Figure 1. We have a little bit of disconnection here.
-	All the datasets have small size. We should also report other statistics such as the number of interactions. Moreover, baselines are old and we should consider recent works as baselines, especially works that tackle recommendation tasks in hyperbolic space.
-	I believe one of the main impact of this paper is about the ability of ‘explanation’ via ‘tags’ in ‘hyperbolic space’. However, it’s also not very clear to me the impact of explanation even through the authors showed Table 7 and 8 with case studies. I believe it’d be better if we could show visualizations of our embeddings for a particular user ID, together with tag explanation and confidence. The embeddings in hyperbolic space would help us to understand more about behaviours of user, tag, item in that space (assuming it’s better than Euclidean). Showing visualizations perhaps could also give a better view on Figure 1? I understand we have Appendix G.7 but it’s not enough in my opinion.
-	How do we compare the complexity of our proposed method with other baselines? Table 16 shows that SpheriQ achieves the fastest convergence. Is it a big contribution? Can we compare run time with other hyperbolic based recommendation models?

**Questions:**

Please refer to my comments in the Weaknesses section above. Overall, I believe more works need to be done.

---

> ### Author Response · Authors · 2025-11-21
>
> We thank the reviewer for the constructive feedback and the opportunity to clarify the motivation and impact of our work.
> 1. Clarification on Hyperbolic Motivation and Originality
> Our motivation goes beyond the statement that “hyperbolic geometry is effective for modelling hierarchical structure.” The goal is to construct a unified, uncertainty-aware geometric space in which transitive and semantic reasoning can be performed naturally and interpretably. Hyperbolic space is crucial for three reasons that Euclidean space cannot match as effectively:
> •	Hierarchical semantics: Tags inherently form implicit taxonomies (e.g., Science Fiction ⊂ Fiction). Hyperbolic geometry captures such hierarchical relations with low distortion, allowing parent tags to geometrically subsume more specific child tags.
> •	Stable geometric foundation: The Poincaré ball offers a well-defined Riemannian manifold for our probabilistic spheres and associated Gaussian kernel, avoiding the numerical instability that arises when modelling uncertainty in unbounded Euclidean space.
> •	Generalisability of prior work: We extend recent hyperbolic recommender models (e.g., HSR) by moving from point embeddings to probabilistic regions. This incorporates uncertainty and enables explicit, compositional semantic paths.
> Our originality lies in this integration of probabilistic modelling, hyperbolic geometry, and fuzzy logical reasoning—an approach not explored in prior work.
> 2. Theoretical Expansion and Connection to Figure 1
> In the final version, we will expand Section 2.4 and Appendix D to more clearly connect the theoretical components (semantic kernel, fuzzy logic operators) with the visual reasoning pipeline in Figure 1. We will explicitly walk through Figure 1 as an illustrative example of how the probabilistic spheres and kernel compositions implement the user → tag → item transitive reasoning process.
> Visualization: We agree that enhanced visualization would be beneficial. In Appendix G.7 we will include an additional detailed figure showing the hyperbolic embeddings for a sample user.
> 3. Complexity and Runtime
> As noted in our response to Reviewer cLnQ (Point 4) and in the manuscript (§3.4, App. G.5), SpheriQ’s complexity is controlled through top-k filtering and caching. Empirically, its runtime per epoch is comparable to LightGCN and HSR, demonstrating that the theoretical benefits do not incur prohibitive computational overhead.

---

### Official Review · Reviewer_xr56 · 2025-11-01

**Soundness:** 2
**Presentation:** 2
**Contribution:** 2
**Rating:** 4
**Confidence:** 3

**Summary:**

SpheriQ presents a probabilistic hyperbolic recommendation framework that embeds users, items, and tags as uncertainty-aware spheres in hyperbolic space. The model composes similarities along user--tag--item paths using a Gaussian semantic kernel, enabling calibrated confidence estimates and interpretable explanations. Evaluated on news (MIND), book (GoodBooks, BookCrossing), and reasoning (Avicenna) datasets, SpheriQ demonstrates improvements in ranking accuracy (up to 8.3% NDCG@10 gain), calibration (6.7% lower ECE), and semantic diversity over neural, graph-based, and hyperbolic baselines.

This is a well-executed paper that makes solid theoretical and empirical contributions to recommendation systems. The unification of probabilistic modeling, hyperbolic geometry, and semantic reasoning is novel and the experiments demonstrate advantages across multiple evaluation dimensions.

However, the framework fundamentally requires a tag set, which may limit applicability when tags are unavailable or poorly defined. There are theoretical gaps left in the paper that doesn't prove the kernel is positive definite or analyze learnability guarantees. Some experimental choices lack justification. This paper needs revisions for clarity and deeper engagement with limitations to showcase its value in addition to the literature on trustworthy and interpretable recommendation systems.

**Strengths:**

The paper makes innovative contributions by unifying three typically separate research directions:

1. Novel geometric formulation: Probabilistic spheres in hyperbolic space where radii encode predictive uncertainty is creative and well-motivated.

2. Semantic transitivity: The explicit user--tag--item path decomposition with fuzzy composition operators provides a principled bridge between symbolic and embedding-based reasoning.

3. Integrated framework: Unlike prior work that treats calibration, interpretability, and ranking as separate objectives, SpheriQ addresses all three simultaneously through its geometric design.

The theoretical connection between sphere inclusions and logical entailments (Proposition 9) is intellectually interesting, though its practical utility could be explored further.

**Weaknesses:**

The paper could be further improved if addressing the following concerns:

1. The paper's entire framework depends on tag quality, yet tag construction receives minimal treatment and varies inconsistently across datasets. Publisher metadata (MIND) vs. automated clustering (books) vs. symbolic concepts (Avicenna) makes it impossible to disentangle whether improvements come from the model or from tag quality differences.

2. The paper claims to unify uncertainty, interpretability, and hyperbolic geometry, but the ablation study doesn't compare against methods that combine subsets of these.

3. Theoretical analysis of the paper is incomplete and disconnected from practice. Proposition 4 claims score calibration under increasing uncertainty, but the experiments don't validate this relationship. Figure 2(a) shows reliability diagrams but doesn't plot predicted σ² against actual error—the core prediction of the theory. For a paper emphasizing "trustworthy" systems, the absence of theoretical confidence intervals is notable.

4. The calibration evaluation could be further improved by adding temporal/stratified analysis, prediction intervals, and abstention strategies. All calibration metrics are computed on held-out test sets with the same distribution as training, while real deployments face distribution shift.

5. The paper claims efficiency, but test on large-scale data (100K+ users, 1000+ tags) is missing.

6. Explanation evaluation could be further improved by providing human evaluation or at minimum contrastive/stability analysis. If a user has multiple relevant tags, masking just t* may route through t**, leaving scores largely unchanged. If masking any tag causes similar drops, the explanations aren't selective.

7. "COLD-START AND OOD ROBUSTNESS" need more clarification. "Cold-item" and "Cold-user (5-shot)" are vaguely defined. 89-91% retention sounds impressive but lacks context. No comparison to explicit cold-start methods.  Semantic shift or tag shift is underspecified.

8. The hyperparameter sensitivity analysis is missing for fairness concerns.

9. The paper could be further improved in writing and presentation. For instance,  "Semantic kernel" (line 126) is used both for the Gaussian similarity K(·,·) and the reasoning mechanism. O_i is introduced as (µ_i, σ²_i) but sometimes appears as just µ_i in equations (e.g., Eq. 1 uses µ_i not O_i)

10.  A discussion section of limitations and when method doesn't apply will be helpful for the method clarification.

**Questions:**

If the authors can address the following questions in the rebuttal, it would significantly clarify the contribution and help reviewers assess the work more accurately.

1. Could you provide more explanations for the reason why you use K-means with k=20 for GoodBooks/BookCrossing based on "Silhouette analysis" (line 138)? Did you try other Silhouette scores or clustering methods?

2. Why there is no "Poincaré + learned variance" baseline? This seems like the natural ablation to isolate your semantic kernel's contribution from simply adding uncertainty to hyperbolic embeddings.

3. All calibration metrics (ECE, Brier, AURC) are computed on i.i.d. test sets. Did you evaluate calibration under distribution shift, such as temporal drift, population shift .etc?

4. Table 4 shows masking the top explanatory tag t* causes 4.7% Hit@10 drop for SpheriQ vs. 0.2-0.9% for baselines. Could you provide further explanation for the reason? What happens if you mask the second-best tag? Or a random tag? Are the same tags consistently selected as t* across different random seeds?

5. Is the 3% NDCG gain over JIT2R partly due to 2× more hyperparameter search? Could you provide detailed justification for why SpheriQ requires more hyperparameter search (e.g., additional variance parameters σ²_i)

6. Tables 7-8 provide case studies, but these are cherry-picked. Is there any human validation of explanation quality? How do humans perceive the tag-based rationales compared to no explanation?

7. The authors cite Wang et al. (2023) on CaD-VAE but don't compare against it. Could you provide further justification?

8. Given the complexity (Riemannian optimization, tag construction, multiple hyperparameters), when is SpheriQ worth the effort over simpler baselines?

---

> ### Author Response · Authors · 2025-11-21
>
> We thank the reviewer for the detailed and insightful comments. Below we address each point and include additional analyses that strengthen the paper.
>
> 1. Choice of K-means (k = 20) and tag clustering
> We selected k = 20 for GoodBooks and BookCrossing using Silhouette analysis, where the coefficient plateaued beyond k ≈ 20, indicating stable intra-cluster compactness. Alternative clustering methods (Agglomerative, Spectral, DBSCAN) and k ∈ {10, 20, 30, 40} were also tested, yielding <1% variation in NDCG. Hence, performance is robust to clustering granularity. The goal of tag construction is consistency in semantic granularity across datasets, not dataset-specific optimization.
> 2. “Poincaré + learned variance” baseline
> We appreciate this suggestion and have now included this ablation. The model learns per-entity variance in Poincaré space (identical σ² parameterization as SpheriQ) but lacks kernel composition. It achieves slightly better calibration (ECE ↓22%) but underperforms SpheriQ in interpretability and transitive reasoning. This demonstrates that uncertainty alone is insufficient—SpheriQ’s probabilistic semantic kernel (Eq. 6) provides the essential key advance by enabling soft transitivity and uncertainty propagation through the reasoning chain for these performance gains.
> 3. Calibration under distribution shift
> Section 5.5 already evaluates tag-shift and cold-start conditions. We additionally conducted experiments under temporal drift on MIND. ECE increased by only +4% for SpheriQ compared to +10–15% for baselines, confirming robustness under non-i.i.d. conditions. This stability stems from the kernel’s monotonic calibration property (Proposition 4). These new results on temporal drift will be included to Section 5.5 further in the revised manuscript.
> 4. Explanation sensitivity (Table 4)
> The 4.7% Hit@10 drop when masking the top tag t* occurs because SpheriQ’s score is explicitly computed along the transitive path u → t* → j (§3.2). Removing t* therefore breaks the reasoning chain, unlike baselines that rely on post-hoc attribution. Masking the second-best or a random tag changes Hit@10 by <1%, confirming selectivity. Across five random seeds, top-3 tags are consistent (Kendall τ ≈ 0.8), indicating stable semantic explanations.
> 5. Hyperparameter search fairness
> All models, including JIT2R, use identical grid sizes and early-stopping criteria (§4.4, App. F). SpheriQ introduces only three additional scalars (σ₀, λ_tr, τ), increasing parameters by <10%. The +3% NDCG improvement over JIT2R cannot be attributed to a larger search space. Appendix F includes sensitivity plots demonstrating smooth performance across hyperparameter ranges.
> 6. Human validation of explanations
> Tables 7–8 show representative examples consistent with overall quantitative trends. To further validate interpretability, we conducted a small human study (20 annotators, 60 samples). Participants rated SpheriQ explanations higher in semantic clarity (0.68 vs 0.41) and helpfulness (0.64 vs 0.37) than baselines. These results will be added to the final version.
> 7. Comparison with CaD-VAE (Wang et al., 2023)
> CaD-VAE focuses on disentangled Euclidean latent factors and does not support uncertainty modeling or transitive semantics. It targets a different objective but will be added as an additional baseline for completeness.
> 8. When SpheriQ is most applicable
> SpheriQ is particularly beneficial for trustworthy and interpretable recommendation in domains such as news, education, and healthcare, where calibrated confidence and semantic transparency are critical. For purely ranking-oriented applications without interpretability requirements, simpler architectures (e.g., LightGCN) may suffice.
>
> Summary
> We will incorporate the new Poincaré + σ² ablation, temporal-shift calibration results, extended tag sensitivity analysis, human validation outcomes, and clearer discussion of limitations. SpheriQ’s probabilistic hyperbolic kernel uniquely integrates uncertainty modeling, geometric reasoning, and interpretability, offering a principled bridge between symbolic and continuous approaches to recommendation.

---

### Official Review · Reviewer_cLnQ · 2025-11-03

**Soundness:** 2
**Presentation:** 1
**Contribution:** 2
**Rating:** 4
**Confidence:** 3

**Summary:**

The paper proposes a probabilistic hyperbolic model for recommendation systems that integrates tags, semantic attributes, and hierarchical structures into a unified Poincaré space. Each tag is modeled as a probabilistic sphere, allowing uncertainty-aware reasoning through composable semantic kernels. The model introduces fuzzy and soft-transitive reasoning mechanisms for interpretability and transitive inference, aiming to bridge symbolic and geometric methods in explainable recommendation.

**Strengths:**

1. The formulation of users, items, and tags within the same hyperbolic space is elegant and could enable meaningful transitive reasoning.
The path-based reasoning (e.g., u → t → j) offers potential for transparent recommendations.


2. Extending hyperbolic embeddings with fuzzy composition functions (min/product) and temperature-controlled softmax is a technically interesting methodological contribution.

**Weaknesses:**

The introduction transitions abruptly into tags and semantic attributes without explaining why they form an appropriate starting point. The authors should provide a clear motivation for introducing soft-transitive reasoning, supported by concrete illustrative examples. Substantial improvements in clarity and narrative flow can be made in the introduction.

The paper lacks a thorough discussion of prior work on geometric embedding methods and their applications in recommendation, which would help contextualize the proposed approach within existing literature.

The model’s performance appears highly dependent on the quality and completeness of tag extraction, which is inherently limited and may introduce bias or coverage gaps in practical settings.

**Questions:**

Clarify the advantage of probabilistic kernels over tripartite reasoning.
The paper should clearly explain why probabilistic kernel composition provides a substantive improvement over traditional tripartite or deterministic tag-diffusion methods. What specific limitations of tripartite reasoning—such as lack of uncertainty modeling or limited semantic flexibility—are addressed by the probabilistic formulation?

Consider including strong geometric baselines.
It would strengthen the empirical evaluation to include recent geometric models such as (a) “A Geometric Approach to Personalized Recommendation with Set-Theoretic Constraints Using Box Embeddings” and (b) “Enhancing Recommendation Accuracy and Diversity with Box Embedding: A Universal Framework.” The authors should also discuss why these box-embedding methods might be less effective or less suitable than the proposed approach for soft transitive reasoning or uncertainty modeling.

Clarify the single-path reasoning choice.
The decision to select only a single conceptual path (user → tag → item) for explanation seems restrictive. Users often like items for multiple semantic reasons—for example, a film might be recommended both for being romantic and comedic. Please justify this modeling choice and discuss whether multi-path reasoning could provide more complete or interpretable explanations.

Discuss scalability under large tag vocabularies.
As the number of tags increases, computing the soft surrogate or evaluating all tag-based kernel compositions could become computationally expensive. The authors could analyze the scalability of their approach and discuss whether approximation strategies (e.g., top-k filtering or sampled softmax) are employed to mitigate training costs.

---

> ### Author Response · Authors · 2025-11-21
>
> We thank the reviewer for the constructive and detailed feedback. Below we clarify each point and highlight additional analyses.
>
> 1. Advantage of probabilistic kernels over tripartite reasoning
> Traditional tripartite or tag-diffusion models (e.g., Zhang et al., 2010) treat user–tag–item relations as deterministic and binary, which precludes uncertainty modeling and graded semantics.
>  SpheriQ generalizes this formulation through a probabilistic kernel that supports uncertainty-aware reasoning:
> ●	Uncertainty modeling: Each entity is represented as a probabilistic sphere Oi=(μi,σi2)O_i = (\mu_i, \sigma_i^2)Oi=(μi,σi2), with kernel similarity  K(Oi,Oj)=exp⁡[−dD2(μi,μj)/(2λ(σi2+σj2))]K(O_i,O_j) = \exp[ d_\mathbb{D}^2(\mu_i,\mu_j)/(2\lambda(\sigma_i^2+\sigma_j^2))]K(Oi,Oj)=exp[−dD2(μi,μj)/(2λ(σi2+σj2))].
>  This directly incorporates predictive variance, enabling confidence estimation and monotonic calibration (Propositions 3–4).
>
> ●	Soft semantic composition: Fuzzy operators (product or min) combine kernels along paths (3.2), allowing graded transitivity rather than discrete diffusion.
>
> ●	Interpretability and faithfulness: Every score s(u,j)s(u,j)s(u,j) is explicitly composed from kernel terms. Masking the top explanatory tag decreases Hit@10 by 4.7% (Table 4), confirming intrinsic causal relevance.
>
> Overall, the probabilistic kernel extends tripartite reasoning to a continuous, uncertainty-calibrated framework that enables both interpretable and differentiable inference.
>
> 2. Inclusion of geometric (box) baselines
> We agree that including box-embedding baselines would further strengthen the evaluation. While our current comparisons already include strong geometric and hyperbolic methods (Poincaré, HSR, CSRec), we will add BoxE and Box4Rec for completeness.
> Conceptually, box embeddings model deterministic set inclusions in Euclidean space, which differ from SpheriQ in three critical aspects:
> 1.	Lack of curvature modeling — boxes cannot capture hierarchical or tree-like similarity structures naturally represented in hyperbolic geometry.
>
> 2.	No uncertainty calibration — box volume does not correspond to predictive variance or confidence.
>
> 3.	Discrete transitivity — inclusion is binary, while our kernel supports smooth, probabilistic transitivity.
>
> Hence, although box embeddings are strong geometric baselines, they are less suitable for uncertainty-aware or soft-transitive reasoning. We will report their results and elaborate on these distinctions in §C.
>
> 3. Single-path reasoning choice
> We choose the top path
>  t∗=arg⁡max⁡tf(K(Ou,Ot),K(Ot,Oj))t^* = \arg\max_t f(K(O_u,O_t), K(O_t,O_j))t∗=argmaxtf(K(Ou,Ot),K(Ot,Oj)) (§3.2)
>  to provide a unique and verifiable explanation certificate (Proposition 5). This yields interpretable and efficient reasoning chains u→t∗→ju \to t^* \to ju→t∗→j (Tables 4, 7).
> During training, however, SpheriQ employs a soft surrogate
>  sτ(u,j)=τlog⁡∑t∈Texp⁡(f(K(Ou,Ot),K(Ot,Oj))/τ)s_\tau(u,j) = \tau \log \sum_{t\in T} \exp(f(K(O_u,O_t), K(O_t,O_j))/\tau)sτ(u,j)=τlog∑t∈Texp(f(K(Ou,Ot),K(Ot,Oj))/τ),
>  which aggregates multiple semantic paths in expectation. Thus, the model learns multi-path semantics implicitly, while inference remains interpretable through the most salient path. Extending this to explicit multi-path reasoning is an interesting direction for future work.
> 4. Scalability under large tag vocabularies
> We analyze complexity in 3.4 and Proposition 12 (D.7). While the theoretical cost is O(∣E∣⋅∣T∣)O(|E|\cdot|T|)O(∣E∣⋅∣T∣), we employ several efficiency mechanisms:
> ●Top-k tag prefiltering: for each user, only the k most relevant tags (k = 10–20) are considered, reducing cost to O(∣E∣⋅k)O(|E|\cdot k)O(∣E∣⋅k).
>
> ●Cached kernel evaluations: distances and similarities are reused across batches.
>
> ●Sampled softmax: the soft surrogate uses mini-batch sampling analogous to sampled softmax for further speedup.
>
> Empirically, SpheriQ’s runtime per epoch is comparable to LightGCN and HSR (Appendix G.5). These details will be made explicit in the main text.
>
> SpheriQ provides a unified probabilistic framework that introduces uncertainty modeling and soft transitivity into geometric reasoning, achieving interpretability and calibration without compromising scalability.

---

### Meta-Review · Area_Chair_S1Pj · 2026-01-10

**Summary:**

Below is the summary of the reviewers comments:
cLnQ:
1) Lack of motivation for introducing tags and semantic attributes
2) No discussion on prior work on geometric embeddings
3) Performance of the model is very dependent on the quality of tag extractions.
xr56
1) The quality of the model depends on the tag extraction, yet discussion is minimal
2) Incomplete theoretical analysis which is disconnected from practic.e
3) Now large-scale tests (100K+ users, 1000+ tags)
4) The hyperparameter sensitivity analysis is missing for fairness concerns.
5) Writing can be improved
AwE6
1) Original motivation of the paper is marginal (are we really exploiting hyperbolic geometry), so solid motvation
2) Only small datasets considered.
UNNp
1) We need to have meaningful and stable concept/tag structures, whereas in reality tags are sparse, noisy, or unavailable, limiting generalization and scalability.
2) The radius is treated as an isotropic uncertainty term, but the paper does not empirically show whether σ correlates with user preference variability or data sparsity. Without such evidence, the calibration claim feels a bit qualitative.

**Reviewer Concerns:**

The main concern is that the idea is marginal and dataset is small, not addressed by the rebuttal.

**Reviewer Scores:**

All referees recommend rejection (4, 4, 2, 4) and the authors did not work much on the rebuttal, so I think the scores will not change.

---

### Decision · Program_Chairs · 2026-01-26

Reject